# Retrieval augmented generation for large language models in healthcare: A systematic review

**Lameck Mbangula Amugongo**[1]*, **Pietro Mascheroni**[1], **Steven Brooks**[2],
**Stefan Doering**[1], **Jan Seidel**[1]

**1** Biostatistics and Data Sciences Department, Boehringer Ingelheim Pharma GmbH & Co. KG, Biberach an der Riß, Germany, **2** Biostatistics and Data Sciences Department, Boehringer Ingelheim (China) Investment Co., Ltd., Shanghai, China

☯ These authors contributed equally to this work.
* mbangula_lameck.amugongo@boehringer-ingelheim.com

**Data availability statement:** All data used in this study are made available as supporting documents. These documents can be accessed and we have attached them as supplementary materials.

## Abstract

Large Language Models (LLMs) have demonstrated promising capabilities to solve complex tasks in critical sectors such as healthcare. However, LLMs are limited by their training data which is often outdated, the tendency to generate inaccurate ("hallucinated") content and a lack of transparency in the content they generate. To address these limitations, retrieval augmented generation (RAG) grounds the responses of LLMs by exposing them to external knowledge sources. However, in the healthcare domain there is currently a lack of systematic understanding of which datasets, RAG methodologies and evaluation frameworks are available. This review aims to bridge this gap by assessing RAG-based approaches employed by LLMs in healthcare, focusing on the different steps of retrieval, augmentation and generation. Additionally, we identify the limitations, strengths and gaps in the existing literature. Our synthesis shows that 78.9% of studies used English datasets and 21.1% of the datasets are in Chinese. We find that a range of techniques are employed RAG-based LLMs in healthcare, including Naive RAG, Advanced RAG, and Modular RAG. Surprisingly, proprietary models such as GPT-3.5/4 are the most used for RAG applications in healthcare. We find that there is a lack of standardised evaluation frameworks for RAG-based applications. In addition, the majority of the studies do not assess or address ethical considerations related to RAG in healthcare. It is important to account for ethical challenges that are inherent when AI systems are implemented in the clinical setting. Lastly, we highlight the need for further research and development to ensure responsible and effective adoption of RAG in the medical domain.

**Funding:** The author(s) received no specific funding for this work.

**Competing interests:** I have read the journal's policy and the authors of this manuscript have the following competing interests: LMA, PM, SD and JS are employed by the by Boehringer Ingelheim Pharma GmbH & Co. KG, SB is an employee of Boehringer Ingelheim (China) Investment Co., Ltd.

## Author summary

Large language models (LLMs), a type of AI that generate content, has shown promises to solve complex problems, but they have some limitations. For e.g., they sometimes generate inaccurate content, and it is not always clear how they come up with their responses. To tackle these issues and ground the responses of LLMs retrieval augmented generation (RAG) has been proposed. This method ground models by providing them with information from external sources. However, we noticed that there is not enough understanding about the best ways to use RAG in healthcare. We conducted a review to gain a deeper understanding of methods, dataset used for RAG and techniques to assess RAG-based LLMs in medical domain. We found that most studies use English or Chinese datasets and that there is a variety of techniques being used. Interestingly, proprietary models like GPT-3.5/4 are the most used. However, there is a lack of standard techniques to evaluate these applications and many studies do not consider the ethical implications of using AI in healthcare. We believe it is crucial to address these issues to ensure that AI can be responsibly and effectively used in healthcare. Our work is a step towards understanding and improving the use of RAG in this important field.

## Introduction

Large Language Models (LLMs) have revolutionised natural language processing (NLP) tasks in various domains, including healthcare. For example, models such as Generative Pre-trained Transformers (GPT) [1,2], LLaMA [3] and Gemini [4], have shown impressive capabilities in generating coherent and contextually relevant text. However, their application in healthcare is hampered by critical limitations, such as the propensity to generate inaccurate or nonsensical information [5]. This issue is often referred to as "model hallucinations" [6] and methodologies for its mitigation are still an active area of research [7].

In healthcare, several LLMs has been customised to aid in different medical tasks. Models such as BioBERT [8] and ClinicalBERT [9] have been proposed, leveraging the power of Bidirectional Encoder Representations from Transformers (BERT) [10]. These models are developed through fine-tuning using biomedical texts with the aim to improve contextual language comprehension within the medical domain. However, they occasionally encounter challenges when dealing with contextual data. To address this contextual need in medicine, Med-Palm was introduced demonstrating good performance in retrieving clinical knowledge and excelling in decision-making on several clinical tasks [11]. However, Med-Palm could not outperform human clinicians, generated bias and returned harmful answers.

To address the aforementioned limitations, a novel approach called Retrieval-Augmented Generation (RAG) was proposed to expose the model to external knowledge sources [12]. RAG combines the power of LLMs with the ability to retrieve relevant information from external knowledge sources, such as medical databases, literature repositories, or expert systems. Briefly, the RAG process involves retrieving relevant information from the knowledge source, and then using the relevant information to generate a response to answer the question. By incorporating a retrieval step, RAG leverages on in-context learning [13] to reduce hallucinations and enhance the transparency of the sources from which the LLM completion is generated. This is particularly important in healthcare, a knowledge-intensive domain that requires accurate, up-to-date, and domain-specific information [14]. In addition, by incorporating up-to-date clinical data and reliable medical sources such as clinical guidelines into

LLMs, the latter can offer more personalised patient advice, quicker diagnostic and treatment suggestions and significantly enhance patient outcomes [15].

Despite the growth in RAG related research, we only came across a few review studies that outline the state-of-the-art in RAG [16] and methodologies for retrievers and generators [17]. To the best of our knowledge, there is no comprehensive review on methodologies and application of RAG for LLMs in the healthcare domain.

This review aims to fill this knowledge gap, providing a systematic analysis of RAG techniques in the medical setting. We examine different architectures and evaluation frameworks, and explore the potential benefits and challenges associated with the integration of retrieval-based methods. Finally, we propose future research directions and comment on open issues of current RAG implementations. Our contributions:

1. Provide a systematic review of RAG-based methodologies applied in the medical domain. Therefore contextualising the scope of RAG approaches in healthcare.
2. Provide an overview of evaluation methods, including metrics used to evaluate the performance of RAG pipelines.
3. Discuss ethical concerns associated with RAG pipelines in critical sectors such as healthcare.
4. Provide insights for future research directions in RAG-based applications.

## Methods

Many researchers have proposed RAG as a way to provide LLMs with up-to-date and user-specific information, not available as part of the LLM's pre-trained knowledge (also known as "grounding the LLM") [18–24]. Our goal is to integrate the existing literature and evaluate the state-of-the-art (SOTA) RAG techniques used in healthcare. Thus, conducting a systematic literature review is a promising method to explore our objective. Moreover, our aim is to enhance the current knowledge about RAG for LLMs in healthcare. We intend to achieve this by employing a systematic and transparent methodology that produces reproducible results. For this, we employed the Preferred Reporting Items for Systematic reviews and Meta-Analyses (PRISMA) [25]. All studies included met the following inclusion criteria:

### Inclusion criteria

- Language of articles: English.
- Articles published between January 2020 and February 2025.
- Articles covers: RAG and LLMs in the medical domain.

The initial criteria used to identify articles, includes 1) only articles available in the English language; 2) articles published between January 2020 and February 2025, including archives with an arxivid or medrxiv id; 3) only papers proposing RAG-based methods applied in the medical domain. Articles that did not meet the above criteria were excluded. In addition, we also excluded articles that met the criteria below.

### Exclusion criteria

- Review articles, including surveys, comprehensive reviews or systematic reviews.
- Papers for which the full text is not available.
- Short conference papers.

## Search technique

First, we carried out a scoping review using Google Scholar and PubMed to identify and retrieve articles that proposed the application of RAG in the medical domain. The fields considered in the search included the title, abstract, and the article itself. The search terms used are available in Table 1. We used specific search terms to retrieve more relevant articles.

## Study selection

Identified papers before screening can be found in the supporting file, S1 File. After identification, the articles were imported in the ReadCube (Digital Science & Research Solutions Inc, Cambridge, MA 02139, USA) literature management software to create a database of references. We used a three-step process for the selection of articles to be included in this study: relevance of the title, relevance of the abstract and finally relevance of the full-text [26]. This process ensured that only papers that met our eligibility criteria were reviewed. Fig 1 illustrates the process used for screening and determining the eligibility and exclusion criteria.

## Data extraction

For each of the selected studies, we extracted the following information: 1) LLMs, 2) embedding (a numerical representation of information), 3) pre-retrieval, 4) post-retrieval, 5) advanced methodologies and 6) outcomes. In addition, we critically evaluated the technique used to assess the performance of the RAG-based application in the medical domain, including ethical concerns such as privacy, safety, robustness, bias, and trust (explainability/interpretability). Screening and data extraction were carried out by multiple reviewers. Lastly, we performed analyses on data extracted from the papers surveyed in this study. Extracted data are provided in supplementary files: S2 File and S3 File.

# Results

## Included studies

We selected and included 70 studies between 2020–2025. The articles were selected from 2,139 articles retrieved from Google Scholar and PubMed after multiple exclusion steps. Lastly, the compressive review includes studies that employed innovative RAG-based approaches addressing questions such as "what information to retrieve", "when to retrieve" and "how to use the retrieved information" to ground LLMs in the medical domain.

## Datasets

Relevant knowledge sources are essential for LLMs to generate clinically correct responses to users' queries. Several datasets have been proposed in the literature to augment the responses

**Table 1. The keywords used to query the selected databases.**

| Database | Search keywords |
|---|---|
| PubMed | (large language models OR LLMs OR "transformer models" OR "Generative AI") AND (healthcare OR medicine OR medical) AND (retrieval OR augmented OR generation OR grounded) |
| Google Scholar | ("Large Language Models" OR "LLMs OR Transformer Models" OR "Generative Models") AND (Retrieval-Augmented Generation OR grounding) AND (healthcare OR medical OR medicine) |

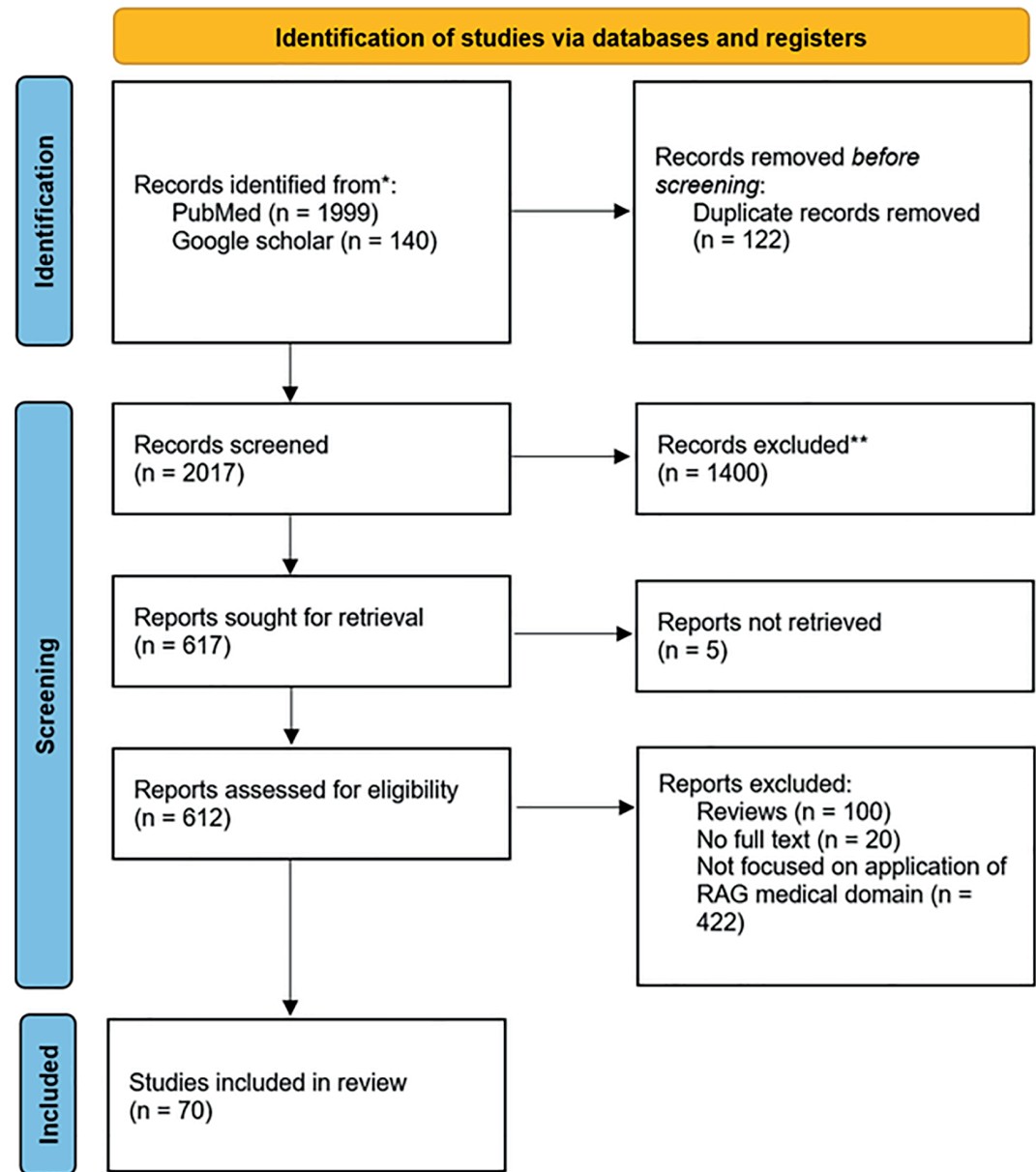

**Fig 1. PRISMA [25] workflow applied to the search, identification and selection of the studies that were included in the systematic review.**

of LLMs in healthcare. Generally, existing retrieval datasets are divided into two categories: question-answering (QA) and information retrieval. As seen in Table 2, most of the datasets used in the studies we reviewed are aimed at QA for medical dialogue. QA dataset provide a short concise answers. On the other hand, information retrieval datasets are geared towards extracting and presenting relevant information to the user's query (generally extracted from large datasets) [27]. The major sources of the dataset used in the studies we surveyed include Web, PubMed, and Unified Medical Language System (UMLS).

Among the available datasets, MedDialog [28] contains conversations and utterances between patients and doctors in both English and Chinese, sourced from two websites,

**Table 2. Detailed information datasets used in retrieval to augment responses of LLMs in the medical domain.**

| Category | Author | Domain | Dataset | #Q in Train | #label in train | #q in dev | #q in test | #instance |
|---|---|---|---|---|---|---|---|---|
| Question Answering | Tsatsaronis et al. [33] | Biomedical | BioASQ | 3,743 | 35,285 | | 497 | 15,559,157 |
| | Chen et al. [28] | Clinical | MedDialog | | | | | EN: 257,454 C: 1,145,231 |
| | Liu et al. [29] | Gastrointestinal | MedDG | 14,864 | - | 2,000 | 1,000 | |
| | Abacha et al. [37] | Biomedical | LiveQA | 634 | - | - | 104 | |
| | Zakka et al. [24] | Biomedical | ClinicalQA | | | | 130 | |
| | Lozano et al. [43] | Biomedical | PubMedRS-200 | | | | 200 | |
| | Jin et al. [35] | Biomedical | PubMedQA | | | 1,000 | 1,000 | |
| | Jin et al. [36] | Biomedical | MedQA | USMLE 10,178 MCMLE 27,400 TWMLE 11,298 | | 1272 3425 1412 | 1273 3426 1413 | 12,723 34,251 14,123 |
| | Ma et al. [44] | Orthodontic | MD-QA | | | | | 59,642 |
| | Chen et al. [41] | 10 pediatric diseases | IMCS-21 | - | - | - | - | 4,116 |
| | Zeng [45] | Biomedical | MMCU-Medical | | | | | 2,819 |
| | Xiong et al. [40] | Biomedical | MEDRAG | | | | | 7,663 |
| | Alonso et al. [42] | Biomedical | MedExpQA | | | | | 622 documents per language |
| Information Retrieval | Boteva et al. [46] | Biomedical | NFCorpus | 5,922 | 110,575 | 24 | 323 | 3,633 |
| | Roberts et al. [31] | COVID-19 | TREC-COVID-19 | - | - | - | - | |
| | Johnson et al. [47] | Radiology | MIMIC-CXR | - | - | - | - | Img: 377,110 Txt: 227,927 |
| | Ramesh et al. [30] | Radiology | Adapted MIMIC-CXR | - | - | - | - | 226,759 |

*Abbreviations: Biomedical semantic indexing and Question Answering (BioASQ), Chinese (C), English (EN), Images (Img), United States Medical Licensing Examination (USMLE), Text (Txt). #q represents the number of queries and #instance the number of texts in the dataset.

namely healthcaremagic.com and iclinic.com. MedDialog dataset covers 96 different diseases. In another study, a name entity dataset called MedDG was created from 17,000 conversations related to 12 types of common gastrointestinal diseases, collected from an online health consultation community [29]. Besides QA datasets, some researchers have curated information-retrieval datasets.

For example, [30] curated a dataset containing 226,759 reports derived from the MIMIC-CXR dataset of X-ray images and radiology report text. Another retrieval dataset is the TREC-COVID-19 dataset, curated using data from medical library searches, MedlinePlus logs, and Twitter posts on COVID by high-profile researchers [31]. Guo et al. [32] presented a dataset linking scientific abstracts to expert-authored lay language summaries. The dataset, named CELLS, was generated with the aim of improving lay language generation models and includes 62,886 source–target pairs from 12 journals.

Benchmark datasets have been released to evaluate the retrieval abilities of LLMs. The Biomedical QA (BioASQ) dataset was proposed through a series of competitions as a benchmark dataset to assess systems and methodologies for large scale medical semantic indexing and QA tasks [33]. Other evaluation datasets for QA tasks include MedMCQA [34], PubMedQA [35] and MedQA [36]. Benchmark datasets such as MedMCQA, PubMedQA and alike do not include broad medical knowledge, and thus lack the detail required for real world clinical applications. To address this limitation, MultiMedQA was created incorporating seven medical QA datasets, including six existing datasets, namely: MedQA, MedMCQA, PubMedQA, LiveQA [37], MedicationQA [38] and MMLU [39] clinical topics, and a new dataset comprising of the most searched medical questions on the internet.

Although MultiMedQA is a useful benchmark, it does not capture the actual clinical scenarios and workflows followed by clinicians. To address this limitation, Zakka et al. [24] curated ClinicalQA as a benchmark dataset containing open-ended questions for different medical specialties, including treatment guidelines recommendations. The authors in [40] introduce MIRAGE, a benchmark for biomedical settings consisting of five commonly used datasets for medical QA. In particular, MMLU-Med, MedQA-US and MedMCQA are included to represent examination QA datasets, while PubMedQA and BioASQ account for research QA settings. Note that all tasks in MIRAGE consist of multi-choice questions, and accuracy and its standard deviation are the default evaluation metrics.

The majority of the existing datasets have insufficient medical labels. For instance, most datasets only provide a single label, e.g. medical entities, which are not detailed enough to represent the condition and intention of a patient in the real world. Moreover, the existing annotated datasets are limited in scale, usually consisting of only a few hundred dialogues [41]. To address these issues, Chen et al. [41] proposed an extensive medical dialogue dataset with multi-level fine-grained annotations comprising five separate tasks which include named entity recognition (NER), dialogue act classification, symptom label inference, medical report generation, and diagnosis-oriented dialogue policy.

Most datasets released to date are curated in English, making it difficult to obtain a non-English dataset. However, recently Chinese datasets such MedDG [29] and MedDialog-CN [28] have been proposed. Retrieval Augmented Generation techniques are designed to solve two main problems: the lack of up-to-date medical information and the tendency of these models to make things up [40]. MEDRAG was proposed as one of the first datasets including 7663 questions for medical QA. Using Medical Information Retrieval-Augmented Generation Evaluation (MIRAGE), showed clear improvements without any prior training on two out of five datasets in their MIRAGE test, while the improvements on the other datasets are smaller. Still, MEDRAG is a useful way to make Medical QA better by adding external medical knowledge [40]. Building on MEDRAG, Medical Explanation-based Question Answering (MedExpQA) a multilingual benchamark for Medical QA was proposed [42]. Unlike previous benchmark datasets for QA, MedExpQA also provides clear explanations for why the correct answers are right and why the other options are wrong. These detailed explanations, written by medical doctors, help evaluate the model's decisions using complex medical reasoning.

## RAG overview

RAG aims to ground the responses of LLMs to provide more factual and truthful responses and reduce hallucination. This is achieved by including new knowledge from external sources. As illustrated in the example of Fig 2, a user asks a question about a new COVID-19 variant ("Tell me about the new KP.3 COVID variant that is dominant in the US: What are the symptoms? The new "FLiRT" COVID-19 variants, including KP.2 and KP.3, are on the rise in the US. Experts discuss symptoms, transmission and vaccines."). An LLM such as ChatGPT will not be able to provide information on recent events because responses from LLMs are time-constrained by the data they are trained on (which is, in the best cases, a few months old). RAG helps LLMs overcome this limitation by retrieving information from up-to-date knowledge sources. In our case, the search algorithm will retrieve a collection of articles related to the prompted virus. Then, retrieved articles together with the prompt are used by the LLM model to generate an informed response.

As seen in Fig 2, RAG workflow comprises three important steps. The first step involves splitting documents into distinct segments, and vector indices are created using an encoder

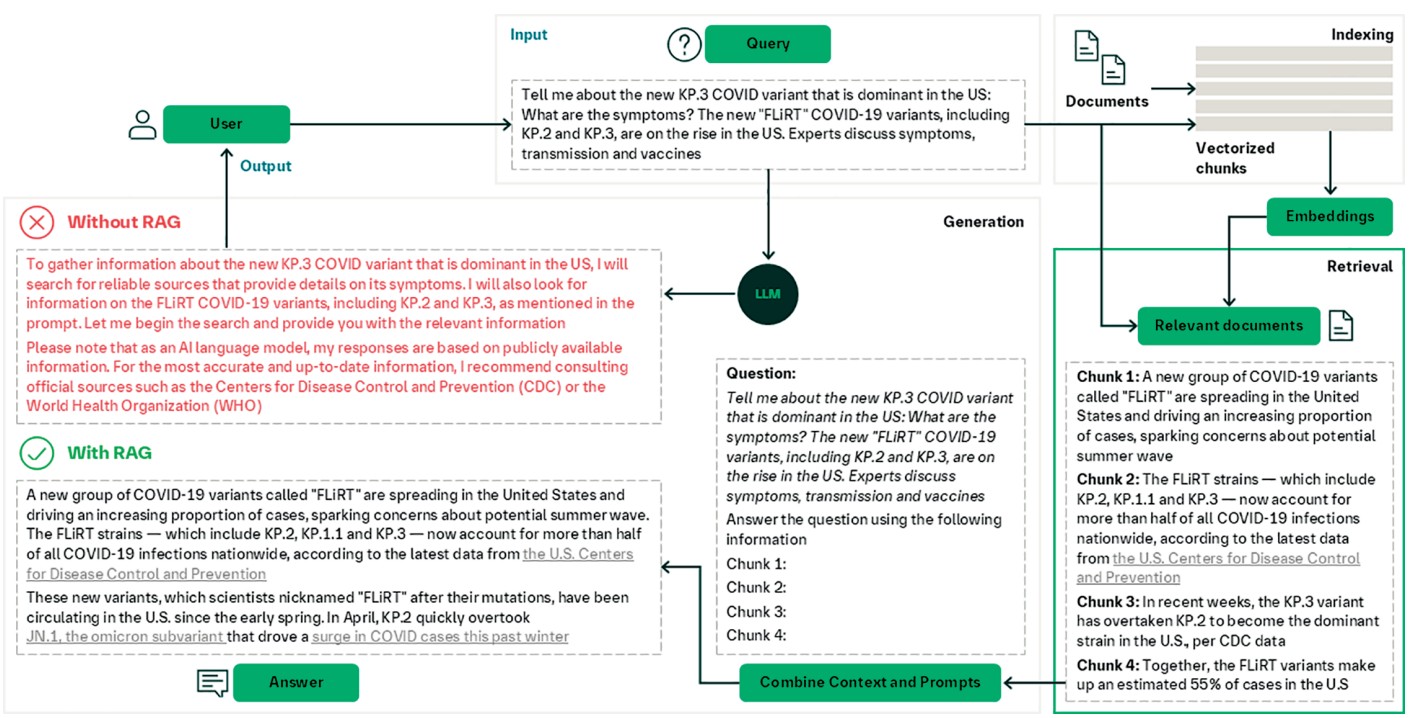

**Fig 2. A schematic illustration of the basic RAG workflow applied to answering a user question in the medical domain.**

model (in the case of dense retrieval). This process is called indexing. After that, segments are located and retrieved based on their vector similarity with the query and indexed segments. The last step involves the model generating a response based on the context derived from the fetched segments and query. These steps constitute the core structure of the RAG process, reinforcing its capabilities in information retrieval and context-aware response generation. RAG leverages a synergistic approach that integrates information retrieval and in-context learning to enhance the performance of an LLM. With RAG, the LLM performance is contextually bolstered without performing computationally expensive model retraining or fine-tuning. This nuanced approach makes RAG practical and relatively easy to implement, making it a popular choice for building conversational AI tools, especially in critical sectors such as healthcare. A recent review outlined the progression of RAG technologies and their impact on tasks requiring extensive knowledge [16]. Three developmental approaches within the RAG framework were outlined: Naive, Advanced, and Modular RAG, each representing a successive improvement over the one before. In the context of this review, we discuss the RAG techniques used in healthcare by grouping them into these three categories.

## Naive RAG

Naive RAG is a basic implementation of RAG and one of the earliest techniques employed to ground LLMs to generate text using relevant information retrieved from external knowledge sources [16]. It is the simplest form of RAG without the sophistication needed to handle complex queries. The Naive RAG process follows the mentioned three steps: indexing, retrieval and generation. Implementations of Naive RAG in the medical domain are discussed below.

Ge et al. [48] used text embeddings to transform guidelines and guidance documents for liver diseases and conditions. Then, they converted user queries into embeddings in real-time

using the text-embedding-ada-002 model and performed a search over a vector database to find matches for the embeddings before generating a response using GPT-3.5-turbo or GPT-4-32k. Their results show that they were able to generate more specific answers compared to general ChatGPT using GPT-3.5.

In another study, [49] presented ChatENT, a platform for question and answer over otolaryngology–head and neck surgery data. For the development of ChatENT, the authors curated a knowledge source from open source access and indexed it in a vector database. Then, they implemented a Naive RAG architecture using GPT-4 as the LLM to generate responses. With ChatENT they demonstrated consistent responses and fewer hallucinations compared to ChatGPT4.0. Zakka et al. [24] proposed Almanac, an LLM augmented with clinical knowledge from a vector database. They used a dataset containing 130 clinical cases curated by 5 experts with different specialties and certifications. The results showed that Almanac provided more accurate, complete, factual and safe responses to clinical questions. In another study, GPT-3.5 and GPT-4 models were compared to a custom retrieval-augmentation (RetA) LLM [50]. The results showed that both GPT-3.5 and GPT-4 generated more hallucinated responses than the RetA model in all 19 cases used for evaluation.

Thompson et al. [51] implemented a pipeline for zero-shot disease phenotyping over a collection of electronic health records (EHRs). Their method consisted in enriching the context of a PaLM 2-based LLM by retrieving text snippets from the patients' clinical records. The retrieval step was performed using regular expressions (regex) generated with the support of an expert physician. Then, a MapReduce technique was implemented to supply the final query to the LLM using only a selection of the complete snippet sets retrieved by the regex. The authors tested several prompting strategies and were able to obtain a improved performance in pulmonary hypertension phenotyping compared to the decision-rule approach devised by expert physicians (F1-score = 0.75 vs F1-score = 0.62, respectively).

The authors in [40] performed a systematic evaluation of naive RAG over a set of medical QA datasets from both examination and research areas. They compared the output of chain-of-thought prompting with and without external context from 5 different medical and general corpora. Inclusion of external knowledge increased GPT-4 average accuracy over multi-choice questions from 73.44% to 79.97%, whereas the average accuracy of GPT-3.5 and Mixtral were improved from 60.69% to 71.57% and from 61.42% to 69.48%, respectively. For questions in which related references can be found in PubMed, RAG strongly improved the performance of LLaMA2-70B (from 42.20% to 50.40%), leading to an accuracy that is close to the corresponding fine-tuned model over the medical domain. In addition, it was shown that by combining different retrievers using rank fusion algorithms leads to improved performances across various medical QA tasks. The authors investigated the impact of the number of retrieved chunks on the accuracy of the responses. They found that for accuracy of response, the optimal number of retrieved chunks varies with the QA dataset: for PubMed-based QA datasets, fewer chunks provided better accuracy, while for examination-based QA datasets, more retrieved chunks the LLM provided a better response.

Guo et al. [32] presented work in which text-to-summary generation is performed. They evaluated the ability of language models to generate lay summaries from scientific abstracts. The authors proposed a series of custom-made language models based on the BART architecture, enriching the text generation with retrieved information from external knowledge sources. They made use of both dense retrieval and definition-based retrieval, with corpora including PubMed and Wikipedia. Finally, the authors compared the output of the custom models to general pre-trained LLMs such as GPT-4 and LLaMA-2. They showed a trade-off between the integration of external information and understandability of the generated output, and validated their observations with both automated and human-based evaluation.

In general, the reviewed literature shows that Naive RAG systems perform better than foundational models alone. Different authors emphasize the importance to tune parameters such as the number of chunks and the retrieval mode to improve the model outcomes. Also, it is found that some Naive RAG approaches may be limited by low retrieval quality, redundancy, and generation errors. To overcome these challenges, advanced RAG methods have been proposed, which use techniques such as chunk optimisation, metadata integration, indexing structure, and context fusion. A review of the latter approaches is provided in the next section.

## Advanced RAG

Advanced RAG is an improvement of Naïve RAG by incorporating additional pre and post retrieval steps to refine information from the knowledge sources [16]. These additional stages improve the quality and accuracy of the information retrieved, ensuring it blends seamlessly with the model's output. This is achieved by employing specialized pre-retrieval and post-retrieval mechanisms with the aim to address challenges faced by Naive RAG such as failure to retrieve all relevant information, problem of integrating context from chunks retrieved and generating answers using irrelevant context. This is achieved by employing different strategies, including pre-retrieval, retrieval and post-retrieval. Pre-retrieval methods aim to optimize data indexing and can be achieved using different strategies such as improving data granularity, optimizing indexing structures, adding metadata, optimizing alignment and using mixed retrieval [16].

During the retrieval phase, dense retrieval-based RAG systems use embedding models to identify suitable context by computing the similarity between the prompt and the chunks. The retrieval step can be optimized by filtering the retrieved chunks using a threshold on their similarity with the user query. For example, Quidwai and Lagana [93] developed a system with a sophisticated retrieval component which allowed for the use of a predefined threshold to determine when insufficient relevant information were retrieved, after which the system can respond "Sorry, I could not find relevant information to complete your request." In this way, they reduced the generation of misleading or false information.

Presenting all retrieved content can introduce noise, shift attention from important content and may exceed the context window limit (number of tokens or words) of the LLM. To overcome the context window limits of LLMs and focus on crucial information, the post-retrieval process combines the query with the relevant context from knowledge sources before feeding it to the LLM. An important step is to reorder the retrieved information so that the most relevant content is closer to the prompt. This idea has been applied in frameworks like LlamaIndex [119] and LangChain [120].

Rau et al. [96] used LlamaIndex and GPT-3.5 to create a context aware chatbot grounded in a specialised knowledge base containing vectorised American College of Radiology (ACR) appropriateness criteria documents and compared its performance to GPT-3.5 and GPT-4. Out of 50 case files, they found that their context-based accGPT gave the most accurate and consistent advice that matched the ACR criteria for "usually appropriate" imaging decisions, in contrast to generic chatbots and radiology experts. Similarly, Russe et al. [97] used LlamaIndex as an interface between the external knowledge and a context-aware LLM (FraC-Chat). They extracted text information from radiology documents using the GPTVectorIndex function, which divides the content into smaller chunks (up to 512 tokens), converting chunks into data nodes. The data nodes were then encoded and stored as a dictionary-like data structure and then used in the answer creation. Using 100 radiology cases, FraCChat performed better on classification of fractures compared to generic chatbots, achieved 57% and

83% correct full "Arbeitsgemeinschaft Osteosynthesefragen" codes with GPT-3.5-Turbo and GPT-4 respectively.

A more recent study has shown that "noise" (documents not directly relevant to the query) can impact the performance of RAG systems - some models such LLaMA-2 and Phi-2 perform better when irrelevant documents are positioned far from the query [121]. Yang et al. [122] explored techniques for dealing with noise, namely: Bio-Epidemiology-NER, direct and indirect extraction to identify medical terminology augment an LLM using UMLS knowledge base. Using GPT-3.5 augmented with UMLS information, they created a trustable and explainable medical chatbot supported by factual knowledge. However, the extracted terminologies are not always related to the question asked, producing incomplete answers. Wang, Ma and Chen [105] used a large dataset containing high quality medical textbooks as an external knowledge source, combined with multiple retrievers to improve LLM performance in generating high quality content.

To improve retrieval and reasoning, researchers are investigating incorporating common-sense knowledge graphs (KGs) with dialogue systems in the medical domain. Two conversational models, MedKgConv [123] and MED [124] for medical dialogue generation were proposed, utilising multi-head attention and knowledge-aware neural conversation respectively. MedKgConv uses a BioBERT encoder to encode conversation history, which is then processed through Quick-UMLS to extract knowledge graphs for reasoning, and a BioBERT decoder for response generation. MED, on the other hand, encodes augmented KGs alongside patient conversations using LLMs, enhancing it through medical entity annotation in a semi-supervised manner. Both models demonstrated improved performance on MedDialog and Covid datasets, with MedKgConv showing an increased 3.3% in F1 score and 6.3% in BLEU-4, and MED outperforming BioBERT by 8.5 points on F1 and 9.3 points on BLEU-4. These results underscore the effectiveness of integrating dialogue and graph-based knowledge in generating medical dialogues. The aforementioned KG approaches are implemented for QA tasks, where KGs contain structured information used as context for predicting the answer. As such, they could have limited versatility.

Recent studies have concentrated on developing approaches that integrate KGs to enhance LLMs and RAG, enabling them to generate accurate and reliable medical responses. Unlike the knowledge found in document repositories, KGs provide structured and inferable information, making them more suitable for augmenting LLMs-RAG [125,126]. Soman et al. [100] proposed a context-aware prompt framework that adeptly retrieves biomedical context from the Scalable Precision Medicine Open Knowledge Engine (SPOKE) [127]. Using their KG-RAG approach, the performance of LLaMA-2 was significantly improved, showing a 71% improvement from the baseline. Another study showed that KG-enhanced RAG can effectively retrieve relevant facts from the KG, generate coherent and informative answers, and explain the connections between the genes, diseases, and drugs related to Alzheimer's [85]. A lack of detailed medical and structured information for diagnostic purposes has been reported as a shortcoming for existing medical KGs. MedRAG was proposed as a framework that integrates RAG with a comprehensive diagnostic KG [114]. This combination improved RAG's reasoning ability, allowing it to identify subtle differences in diagnoses with similar manifestations. Furthermore, they have shown that RAG enhanced by KG outperformed naive RAG with chain of thought.

In the realm of clinical development, advanced RAG techniques have been explored for clinical trial patient matching [108,128] and to accurately identify and report on inclusion and exclusion criteria for a clinical trial [103]. For instance, Jin et al. [128] employed aggregated ranking to perform patient-trial-matching using clinical notes. Similarly, a recent study used two retrieval pipelines, first selecting the top-k most pertinent segments from the patients'

notes, and then using top-k segments as a prompt input to assess the LLM [108]. Their results demonstrated that it is possible to reduce processes that typically takes an hour-per-patient to a matter of seconds.

Another study used GPT-4 enabled with clinical notes through RAG for clinical trial screening [103]. The authors used LangChain's recursive chunking to divide patient notes into segments to preserve the context. To optimise the retrieval, they employed Facebook's AI Similarity Search (FAISS) [129]. They showed that using GPT-4 with RAG to screen patients for clinical trials can improve efficiency and reduce costs.

In summary, Advanced RAG systems are shown to improve over using only foundational models or Naive RAG approaches. The reviewed literature highlights a trade-off between complexity and performance gains, with more complex RAG implementations providing better outcomes at the cost of more application blocks to implement and maintain. In addition to that, some approaches stress the importance of ranking of the retrieved chunks, or the use of multiple retrievers to improve extraction of the required information from the knowledge source. With respect to the latter component of RAG, some authors employ particular implementations of the knowledge source (e.g., knowledge graphs). The improvements in response generation compared to more standard approaches depend on the nature of the information included in the knowledge source and the complexity of the user queries.

## Modular RAG

Modular RAG incorporates several techniques and modules from advanced RAG, allowing more customisation and optimization of the RAG system, as well as integration of methods to improve different functions [16]. For example, modular RAG can include a search module for similarity retrieval, which can improve the quality and diversity of the retrieved content, and apply a fine-tuning approach in the retriever, which can adapt the retriever to the specific domain or task [130].

Wang, Ma and Chen [105] presented an LLM augmented by medical knowledge, using modules consisting of hybrid retrievers, query augmentation and an LLM reader. Each module enhanced a particular task, e.g., the query augmentation module improved prompts for effective medical information retrieval and the LLM reader module provided medical context to the question. They reported an improvement in response accuracy ranging between 11.4% to 13.2% on open-medical QA tasks, as compared to the GPT-4-Turbo without RAG. Despite using a smaller dataset, they showed that using medical textbooks as a knowledge source outperformed Wikipedia in the medical domain. This highlights the importance of context and quality information in specialised domains such as healthcare.

A recent study by Jin et al. [69] proposed a framework that integrates advanced techniques such as large-scale feature extraction combined with RAG, accurate scoring of features from medical knowledge using LlamaIndex, and XGBoost [131] algorithm for prediction. By employing this modular approach, they showed improved prediction of potential diseases, surpassing the capabilities of GPT-3.5, GPT-4, and fine tuning LLaMA-2.

In another study, a new approach to perform RAG is presented, eliminating the reliance on vector embedding by employing direct and flexible retrieval using natural language prompts [72]. The authors used an LLM to handle the step of document retrieval to response generation without needing a vector database and indexing, simplifying the RAG process. They showed the performance of prompt-RAG through a QA GPT-based chatbot using Korean medicine documents. They showed that the novel prompt-RAG achieved good results, outperforming ChatGPT and RAG-based models which used traditional vector embedding. Specifically, based on ratings from three doctors, prompt-RAG scored better in relevance and

informativeness, similarly on readability. On the downside, the response time was significantly slower.

When user queries contain limited context information, the retriever may be unable to retrieve relevant documents from the knowledge sources. A method that uses hypothetical outputs generated from user queries has been proposed [18], improving performance in zero-shot scenarios. Compared to knowledge stored in unstructured documents such as in the portable document format (PDF), KGs are more ideal for RAG because of the easiness in accessing relations among knowledge items [132]. Ongoing explorations are focused on designing the best strategy to extract information from KGs and to facilitate interaction between LLMs and KGs. For instance, [68] presented a Hypothesis Knowledge Graph Enhanced (HyKGE) framework to improve the generation of responses from LLMs. The HyKGE comprises 4 modules, namely: hypothesis output, Named Entity Recognition (NER), KG retrieval, and noise knowledge filtering. Experimenting on two medical QA tasks, the authors demonstrated a 4.62% improvement compared to baseline in F1 score with their modular HyKGE framework. HyKGE also was able to address challenges such as poor accuracy and interpretability, and showcased potential application in the field of medicine.

As a general trend, researchers are now exploring the implications of having multiple LLMs jointly working together. For instance, Lozano et al. [43] proposed a RetA LLM system called Clinfo.ai consisting of 4 LLMs, jointly forming an LLM chain. They employed the first LLM to perform an index search on either PubMed or Semantic Scholars, an LLM for relevance classification, an LLM for article summarisation using a user query and the fourth LLM using task-specification prompts to guide the LLM output. Another study proposed a RAG-based method employing multiple LLM agents for feature extraction, prompt preparation and augmented model inference [111]. The authors evaluated their approach using two datasets for diagnosing arrhythmia and sleep apnea. The findings suggest that their zero-shot strategy not only outperforms previous methods that use a few-shot LLM, but also comparable to supervised techniques trained on large datasets. Woo et al. [109] has demostrated that incorporating AI agents into a previously RAG-augmented LLM improved the accuracy of GPT-4 to 95%. Agentic approaches are increasingly being used to improve capabilities of LLMs in solving complex problems by sharing tasks across multiple agents [133]. Multi-agents systems have the potential to improve the capabilities of LLMs in solving complex problems in the medical domain, by splitting tasks into multiple sub-tasks and assigning specific LLM agents to each of them.

In summary, approaches that implement the Modular RAG framework are characterized by more complex pre- and post-retrieval steps, showcasing reformulation of the user query or LLM readers. From the trend in the more recent literature, we envision a wider and wider adoption of agentic frameworks in RAG systems. LLM-based agents [134] allow for a redefinition of complex tasks into simpler ones, with dedicated "reasoning engines" to tackle them. This has the advantage of casting the original problem into a modular approach, with the related pros and cons in terms of managing maintenance and complexity. A recent study showed that a modular, lightweight RAG framework can efficiently tackle complex medical question answering using social media data in low-resource environments [58]. This framework allows clinicians to quickly gather insights on substance use trends and potential side effects from Reddit posts, offering substantial promise for enhancing public health and operationalising LLMs in low-resource settings.

To conclude this section, we provide a summary of the reviewed literature, encompassing the different RAG implementations, in Table 3. Overall, the reviewed implementations of RAG in healthcare follow the three paradigms of RAG, namely Naive, Advanced, and Modular RAG, as described in [16]. Across each RAG implementation, researchers have proposed

**Table 3. A detailed list of different RAG methods used in the surveyed studies.**

| Authors | LLMs | Embedding | Pre-retrieval | Post-retrieval | Adv. Meth. | Outcomes |
|---|---|---|---|---|---|---|
| Abdullahi et al. [52] | ClinicalBERT PubMedBERT SciBERT SapBERT CODER | MedCPT | Chunking | TF-IDF RRF | Zero-shot prompting | CliniqIR outperformed supervised fine-tuned models for diagnoses with <5 training samples |
| Alkhalaf et al. [53] | LLaMA-2-13B | NA | Chunking | SS MMR | Zero-shot prompting | RAG approach improved the model performance and mitigated the hallucination problem |
| Al Ghadban et al. [19] | GPT-4 | NA | Chunking | SS MMR | One-shot learning | Acc: #141 (79%). Adeq: #49 (35%) Promising role RAG and LLMs in medical education |
| Aratesh et al. [54] | GPT-3.5T GPT-4 Mistral-7B Mixtral-8x7B LLaMA-3-8B LLaMA-3-8B | ada-0002 | Chunking | SS | Zero-shot inference | RAG enhanced diagnostic accuracy in most LLMs, relative accuracy improvements reaching up to 54% |
| Azimi et al. [55] | GPT-4o Claude 3.5 Sonnet Gemini Pro 1.5 | Titan Text Embeddings V2 | Chunking | SS | RAP | RAP was particularly effective for GPT-4o to answer Expert level questions |
| Benfenati et al. [56] | GPT-3.5T Mistral-7B | GTE | Chunking | SS | Augmented prompt | RAG improves quality of responses for baseline LLMs |
| Bora and Cuayáhuitl [57] | LLaMA-2-7B Flan-T5-Large Mistral-7B | gtr-t5-large | Chunking Indexing | SS | Few-shot prompting | FT + RAG improved LLM performance |
| Chen et al. [21] | GPT-4 | NA | OIS | SS | Zero-shot CoT | Improved accuracy with RAG compared foundational models |
| Chen et al. [22] | LLaMA-2-7B-chat | all-MiniLM -L6-v2 | Chunking | SS | FT | FT+RAG provided best performance |
| Das et al. [58] | Nous-Hermes-2-7B GPT-4 | | Chunking | RR Summary | | RAG framework can effectively answer medical questions and suitable in resource-constrained settings |
| Duan et al. [59] | GPT-3.5-T GLM-3-T Qwen-T Spark3.5 Max Moonshot-V1-8K | | NER | knowledge fusion | few-shot learning | Achieved faithfulness: 0.9375 relevancy: 0.9686 & recall: 0.9500 |
| Fukushima et al. [60] | JGCLLM | GLuCoSE-base-ja [61] | | SS | Enhanced prompt | RAG improved critical aspects of genetic counseling, outperforming instruction tuning and prompt enginering |
| Gao et al. [62] | T5 [63] GPT-3.5-T | | OIS KGs | Path ranker Aggregated | Zero-shot with path prompts | Improved diagnosis performance |
| Garcia et al. [64] | LLaMA-3-70B | BGE | Metadata | SS RR | | Enhanced precision and reduced "hallucinations" risks. |
| Ge et al. [48] | GPT-3.5-T GPT-4 | ada-002 | EDG | SS | Prompting strategy | 7/10 completely correct with RAG |
| Griewing et al. [65] | Mixtral8x7B | NA | Chunking | Query diversification dual-retrieval | Reranking ensemble retrieval | High concordance with tumor board recommendations |
| Guo et al. [32] | LLaMA-2 GPT-4 | BERT | Alignment optimisation | RR Summary | RALL | Improved generation performance and interpretability |
| Jeong et al. [23] | Self-BioRAG | | Chunking | RR | Critic & LLMS Generator | 7.2% absolute improvement over SOTA with 7B or less |
| Jia et al. [66] | PodGPT | BGE [67] | Chunking | SS RR | two-stage retrieve rerank approach | Shows integrating podcast data to enhance language models |

*(Continued)*

**Table 3.** (Continued).

| Authors | LLMs | Embedding | Pre-retrieval | Post-retrieval | Adv. Meth. | Outcomes |
|---|---|---|---|---|---|---|
| Jiang et al. [68] | GPT-3.5 Baichuan-13B | BGE | OIS KGs | RR | Query expansion CoK Noise filtering | Superior performance with RAG F1: 4.62% better than baseline |
| Jin et al. [69] | GPT-3.5-T GPT-4 | ada-002 | Chunking | SS | Integration with XGBoost | F1: 0.762. Acc: 83.3%. RAG surpasses the performance traditional methods |
| Jin et al. [70] | GPT-3.5-T | | | | In-context learning. Codex | GeneGPT outperformed: new Bing, biomedical LLMs BioMedLM and BioGPT, as well as GPT-3 and ChatGPT on 8 RAG tasks |
| Hou and Zhang [71] | GPT-4 | text-embedding -3 small | KGs | SS | | RAG outperforms standalone LLMs achieving over 95% accuracy |
| Kang et al. [72] | GPT-3.5-T GPT-4 | None | Create TOC | Truncation Summary | Use LLM for retrieval & generation | Improved retrieval capabilities Score: 5.5. |
| Ke et al. [73] | GPT-4 GPT-3.5-T LLaMA-2 7B LLaMA-2 13B | ada-002 | EDG Chunking | SS | | Acc: 91.4%. RAG the performance comparable to human evaluators Faster decision-making |
| Ke et al. [74] | GPT-3.5 GPT-4, GPT-4o LLaMA-2-7B LLaMA-2-13B LLaMA-2-70B LLaMA-3-8B LLaMA-3-70B Gemini-1.5-pro Claude-3-Opus | NA | Chunking Metadata | SS | Auto-Merging Retrieval Prompt engineering | GPT-4 LLM-RAG model achieved the highest accuracy (96.4% vs. 86.6%, p=0.016) |
| Klang et al. [75] | GPT-3.5-T GPT-4 LLaMA-3.1-70B LLaMA-3.1-8B Qwen-2-7B Qwen-2-72B Gemma-2-9B Phi-3.5 | GIST-large-embedding-v0 [76] | | SS | Zero-shot prompting | Performance improvements with RAG-enhanced LLMs |
| Kresevic et al. [77] | GPT-4 | | Chunking | SS | Prompt engineering | RAG + prompt engineering out-performs the baseline LLM in producing accurate guideline-specific recommendations |
| Lee et al. [78] | GPT-4o | ada-002 text-embedding-3-small text-embedding-3-large | Chunking | Ensemble retrieval | tokenisation strategies | RAG system that enhances LLM reli-ability in diabetes management across different languages |
| Li et al. [79] | LLaMA-7B | NA | EDG Chunking | RR | FT + wikipedia retrieval | RAG improved acc & efficiency F1 score: 0.84. Reduced workload |
| Li et al. [80] | GPT-4 | text-embedding-3-small | | | In-context learning | FAVOR-GPT achieved relevance: 0.865 and accuracy: 0.85 compared to regular GPT-4 with relevance: 0.5 and accuracy: 0.595 |
| Long et al. [49] | GPT-4 | ada-002 | Chunking | SS | Knowledge specific database | Improved performance with RAG over base models |
| Long et al. [81] | Self-BioRAG | | Chunking Indexing | SS RR | Few-shot learning | Achieved an average performance improvement of 20.72% |
| Lozano et al. [43] | GPT-3.5-T GPT-4 | NA | Relevance Classifier | Synthesis Summary | Online search | Improvement with RAG over base models |
| Luo et al. [82] | Baichuan-13B | SBERT | Chunking | SS | fine-tuning | Achieved SOTA performance in oph-thalmology with accuracy, utility, and safety |

*(Continued)*

**Table 3**. (Continued).

| Authors | LLMs | Embedding | Pre-retrieval | Post-retrieval | Adv. Meth. | Outcomes |
|---|---|---|---|---|---|---|
| Markey et al. [83] | GPT-4 | NA | Chunking | SS | Online search. | Potential for GenAI-powered clinical writing |
| Mashatian et al. [84] | GPT-4 | | | | Zero-shot and Few-shot learning | RAG achieved 98% accuracy providing reliable medical information |
| Matsumoto et al. [85] | GPT-4 | | | | GoT | KRAGEN outperformed baseline GPT models |
| Murugan et al. [86] | GPT-4 | ada-002 | MR | SS MMR | Prompt engineering Guardrails | Improvements with RAG |
| Neupane et al. [87] | GPT-3.5-T Mistral-7B -Instruct | ada-002 | Structured context Chunking | Contextual compression | Online search | Efficacy in generating relevant responses. GPT-3.5-T: 0.93 & Mistral-7B: 0.92 |
| Ong et al. [88] | GPT-4 Gemini Pro 1.0 Med-PaLM-2 | ada-002 bge-SENv1.5 [89] | Manual indexing Auto-merging retrieval | SS | LLM vs "copilot" | RAG-LLM outperformed LLM alone |
| Painter et al. [90] | GPT-4 | ada-002 | Structured context Indexing Chunking | SS | | RAG framework to improve retrieval for safety data |
| Pang et al. [91] | GPT-3.5 | EM-FT | Indexing | SS | fine-tuned embedding | improves recall and precision compared to using the embedding model directly for retrieval |
| Parmanto et al. [92] | LLaMA-2-7B Falcon-7B GPT 3.5-T | BGE [67] | Chunking | SS | FT | RAG + FT best results |
| Quidwai & Lagana [93] | Mistral-7B -Instruct | bgeSENv1.5 | Indexing Chunking | SS | Pubmed dataset curation | Improved accuracy over base models |
| Ranjit et al. [94] | davinci-003 GPT-3.5-T GPT-4 | ALBEF [95] | | Compression | Coupling to vision model | RAG achieved better outcomes BERTScore: 0.2865. Semb: 0.4026 |
| Rau et al. [96] | GPT-3.5-T GPT-4 | ada-002 | Chunking | SS | Visual interface | Superior performance with RAG Time and cost savings |
| Russe et al. [97] | GPT 3.5-T GPT-4 | ada-002 | Chunking | SS | Prompting strategy | Acc: GPT 3.5-T: 57% GPT-4: 83% |
| Shashikumar et al. [98] | Llama-3 8B Mixtral 8x7B | | Chunking | SS | majority vote | Smaller open-weights models are as effective and more efficient than older generation larger LLMs |
| Shi et al. [99] | GPT-3.5-T | MP-Net | Chunking MR | SS | ReAct architecture | Improved performance with RAG over baseline models |
| Soman et al. [100] | LLaMA-2-13B GPT-3.5-T GPT-4 | PubMedBert MiniLM | KGs | Similarity Context pruning | | KG-RAG enhanced performance |
| So et al. [101] | GPT-3.5-T GPT-4-T | ada-002 | Chunking | SS | Zero-shot learning FT | RAG provide a performance advantage over the zero-shot inference in GPT-4-T |
| Soong et al. [50] | Prometheus GPT-3.5-T GPT-4 | ada-002 | Chunking | Summary | | Improved performance with RAG over baseline models |
| Steybe et al. [102] | GPT-4 | ada-002 | Chunking Adding metadata | SS RR | Filtering Prioritise sources | RAG improved response quality and reliability of LLMs |
| Thompson et al. [51] | Bison-001 | NA | Token splitter. Regex | Map Reduce | Regex + LLM aggregation | RAG-LLM outperformed rule-based method. F1: 0.75 |
| Unlu et al. [103] | GPT-4 | ada-002 | Adding metadata Chunking | SS | Iterative prompting | Potential to improve efficiency and reduce costs. Acc: 92.7% |

*(Continued)*

**Table 3.** (Continued).

| Authors | LLMs | Embedding | Pre-retrieval | Post-retrieval | Adv. Meth. | Outcomes |
|---|---|---|---|---|---|---|
| Vaid et al. [104] | GPT-3.5 GPT-4 Gemini Pro LLaMA-2-70B Mixtral-8x7B | | | | CoT prompting Agents | RAG with GPT-4 achieved best performance |
| Wang et al. [105] | LLaMA-2-13B GPT-3.5-T GPT-4 | | QO HR | Knowledge self-refiner | LLM-aided pre- and post-retrieval | RAG outperform baseline models |
| Wang et al. [106] | LLaMA-2-7B LLaMA-2-13B | ColBERT | QO | RR | JMLR | Demonstrate potential of joint IR & LLM training |
| Wang et al. [107] | GPT-4 Claude-2 Bard | ada-002 | Chunking Summarisation | SS | Query re-writing | Significant improvement LLMs' performance in responding to diabetes-related inquiries enhancing accuracy, comprehensiveness, and understandability |
| Wornow et al. [108] | GPT 3.5-T GPT-4 LLaMA-2-70B Mixtral-8x7B | MiniLM BGE | Chunking | SS | Compared zero-shot and retrieval | RAG with GPT-4 beats SOTA in zero-shot |
| Woo et al. [109] | GPT-4, GPT-3.5 Claude-3 LLaMA-3-8B LLaMA-3-70B Mixtral-8x7B | | | | Agentic | RAG improved the accuracy by an average 39.7% Agents improved RAG accuracy rate to 95% with GPT-4 |
| Wu et al. [110] | LLaMA-2-13B LLaMA-2-70B LLaMA-3-8B LLaMA-3-70B Gemini-1.0-pro GPT-4 | text-embedding-3-large | KGs | SS | U-Retrieval: Top-down precise retrieval and Response Refinement | Consistently outperforms SOTA models across all benchmarks Ensuring that responses include credible source documentation |
| Yu, Guo and Sano [111] | LLaMA-2-7B LLaMA-2-7B GPT-3.5 | ada-002 | Chunking | SS | Feature extraction from ECG | RAG outperform few-shot approach |
| Yu et al. [112] | LLaMA-2 | MiniLM | | SS | FT | FT + RAG improved accuracy of information retrieval |
| Zakka et al. [24] | text-davinci-003 | ada-002 | Chunking | Similarity threshold | Adversarial prompting | RAG-LLM outperform ChatGPT |
| Ziletti and D'Ambrosi [113] | GPT-3.5-T GPT-4-T Gemini Pro 1.0 Claude 2.1 Mixtral-8x7B Mixtral-Medium | bgeSENv1.5 | Entity masking | SS EN | Text-to-SQL | GPT-4-T best accuracy and executability. |
| Zhao et al. [114] | Mixtral-8x7B Qwen-2.5-72B LLaMA-3.1-70B LLaMA-3.1-8B GPT-3.5T GPT-4o-mini GPT-4 | Custom | KGs | Similarity Aggregate | KG-elicited reasoning | MedRAG enhanced by KG outperformed naive RAG |
| Zheng et al. [115] | GPT-4o GPT-4o-mini Mistral-8x7B | FastText [116] SBERT text-embedding-3-large | | SS MMR | | Significantly improved accuracy and patient-physician matching |

*(Continued)*

**Table 3**. (Continued).

| Authors | LLMs | Embedding | Pre-retrieval | Post-retrieval | Adv. Meth. | Outcomes |
|---|---|---|---|---|---|---|
| Zhuo et al. [117] | GPT-3.5T | gte-base-zh [118] | Chunking | SS | FT embedding | RAG improve the accuracy and reliability of LLM |

∗Accu racy (Acc); Advanced Methodologies (Adv. Meth.); OpenAI's text-embedding-ada-002 model (ada-002); ALign the image and text represen-tations BEfore Fusing (ALBEF); Accuracy (ACC); BAAI general embedding (bge); bge-small-en-v1.5 (bgeSENv1.5); Chain-of-Knowledge (CoK); Chain-of-Thought (CoT); Text-davinci-003 (davinci-003); Desnse X Retrieval (DXR), Dense Passage Retriever; Enhancing data granularity (EDG); Entity normalisation (EN); Fine-tuning (FT); Frequency (Freq); Joint Medical LLM and Retrieval Training (JMLR); In-context learning (ICL); Image-Text Contrastive learning (ITC); General Text Embeddings (GTE); Graph-of-thoughts (GoT); Hybrid Retrival (HR); Hypothesis Knowledge Graph Enhanced (HyKGE); Knowledge graphs (KGs); Maximal Marginal Relevance (MMR); all-MiniLM-L6-v2 (MiniLM); Mixed Retrieval (MR); N/A (Not Available); Named Entity Recognition (NER); Retrieval Augmented Prompting (RAP); Retrieval-Augmented Language Modelling (RALM); Retrieval Augmented Generation (RAG); Regular expression (Regex); Re-ranking (RR); Retrieval-Augmented Lay Language (RALL); Optimising Index Structure (OIS); Query Optimisation (QO); Sentence Bidirectional Encoder Representations from Transformers (SBERT); Similarity Search (SS); State-of-the-art (SOTA); Table of Contents (TOC); Turbo (T).

different solutions to improve the retrieval and generation steps. In the next section, we will introduce the common evaluation criteria that are being adopted to assess performance of RAG-based applications.

## Evaluation metrics and frameworks

In a RAG system, there are three important aspects: 1) the user prompt (that is, the user query), 2) the retrieval of the context related to the prompt, and 3) using the retrieved context and the user prompt to produce output. Because there are usually no reference answers to questions (i.e., *ground truth*), evaluations have been focused on quality aspects, separately evaluating the retriever and generator. Commonly used evaluation metrics for RAG-based applications include: 1) **Accuracy/Correctness** - e.g., by comparing generated text to ground truth or using human evaluators to assess the correctness of LLM outputs; 2) **Completeness** - the proportion of all retrieved context that is relevant to the user query; 3) **Faithfulness/Consistency** - the degree to which the LLM's output is grounded in the provided context and factually consistent (i.e., not an hallucination) [135]; 4) **Relevance** - whether the generated answer addresses the question asked by the user (*answer relevance*), and whether the retrieved context is relevant to the query (*context relevance*); 5) **Fluency** - the ability of the system to generate natural and easy-to-read text. The evaluation metrics employed in the surveyed studies are presented in Fig 3.

To date, there is no harmonized evaluation approach, with different frameworks using different metrics. For example, Retrieval Augmented Generation Assessment (RAGAs) [135] is one of the most commonly used frameworks to evaluate RAG-based systems. It evaluates a RAG pipeline using four aspects: faithfulness, answer relevance, context relevance and context recall, combining these four aspects to generate a single score to measure the performance of the system. Other packages, such as continuous-eval, have proposed a combination of deterministic, semantic and LLM-based approaches to evaluate RAG pipelines [136]. In addition, TrueLens framework also provides metrics for objective evaluation of RAG pipelines [137]. Another evaluation framework is DeepEval [138], an open-source framework for evaluating LLMs, including RAG applications.

The previously described Clinfo.ai from Lazano et al. [43] introduced a novel dataset, PubMedRS-200, which contains question-answer pairs derived from systematic reviews. This dataset allows for automatic assessment of LLM performance in a RAG QA system. Their framework and benchmark dataset are openly accessible to promote reproducibility. In another study, a toolkit for evaluation of Naive RAGs was proposed [40]. Briefly, the authors

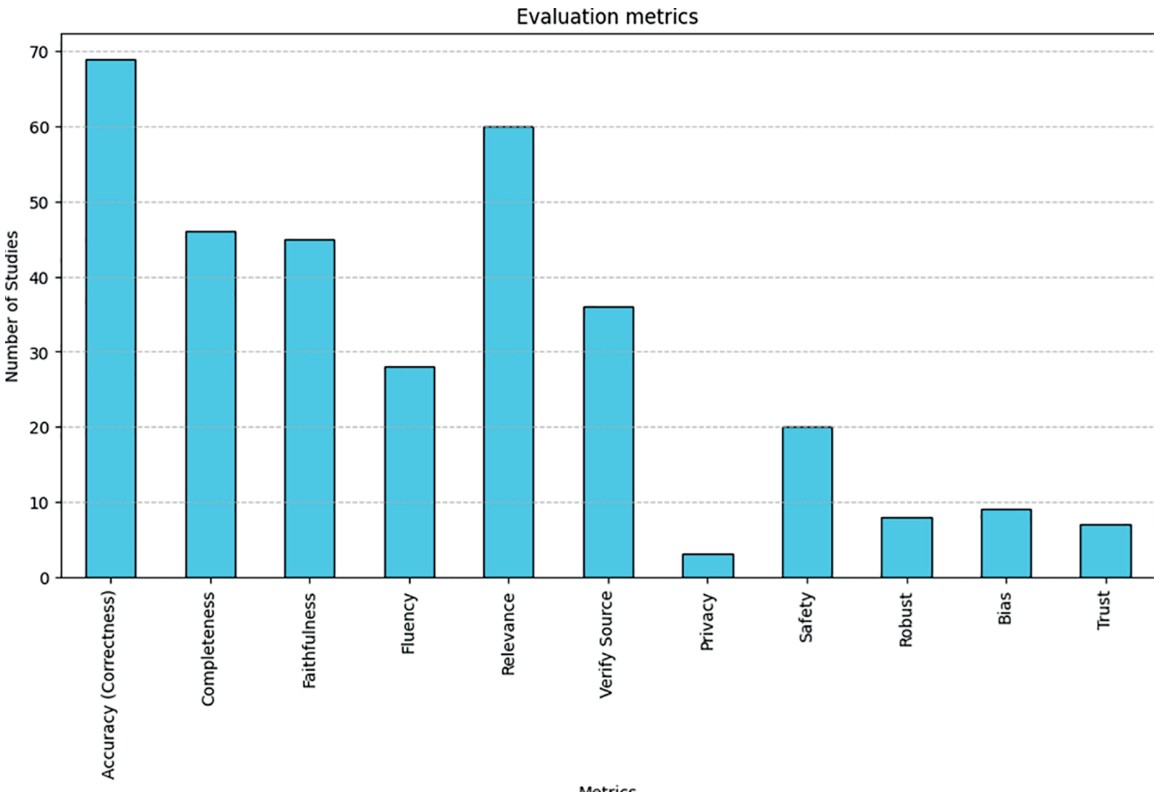

**Fig 3. Metrics used to evaluate RAG-based systems in the medical domain.** Additionally, we assess whether ethical principles: privacy, safety, robustness (robust), bias, trust (explainability/interpretability) have been considered.

considered different LLMs (both from general and biomedical domains), supplemented with five diverse corpora. They compared the performance on medical QA tasks using RAG over four different retrievers. As the tasks in the QA set required the choice between a given set of answers, they chose to use accuracy as the evaluation metric.

Guo et al. [32] presented an evaluation of LLM-generated lay summaries that comprises both automatic and human-based metrics. For evaluating the quality of the synthetic text generation to the target text, they employed ROUGE-L, BERTScore, BLUE and METEOR metrics. Notably, this list of metrics includes both lexicon-based metrics and embeddings similarity. Also, the authors used the Coleman-Liau readability score and word familiarity to assess the easiness of text readability. In addition, the authors trained a RoBERTa model [139] to generate a "Plainness Score", a metric that indicates how much a generated text is representative of the target one. In their effort to holistically evaluate the output of the language models, the authors also employed human reviewers for evaluating characteristics such as grammar, understandability, meaning preservation, correctness of key information and relevance of external information in the generated texts. The human evaluators gave their evaluation on a Likert scale.

Despite the progress achieved in evaluating RAG systems, two main limitations still remain: reliance on human evaluations, which hinders scalability and comparability, and the focus on output quality without considering the relevance of the information used. The challenge of selecting relevant sources for summarisation is as significant as the summarisation

itself. Therefore, a benchmark is needed for comprehensive evaluation of document selection and summary generation capabilities.

In summary, the evaluation landscape of RAG-based applications is still in an early stage, with several library-specific metrics put forward by different teams. As a general remark, there is a need to translate from pure word/gram-based metrics to more complex ones that can capture the generative nature of LLM outputs. Also, there is a suite of benchmarks and metrics developed for information retrieval that should be adapted and refined for the retrieval part of RAG systems.

## Ethical considerations

Healthcare is one of the most regulated industries, guided by principles such as bioethics [140] and various data regulations. For LLMs to be accepted and adopted in healthcare, ethical considerations surrounding their use such as bias, safety and hallucination need to be addressed.

Recently, RAG has shown the ability to boost the performance of LLMs by grounding LLM's responses using retrieved knowledge from external sources. While the benefits of RAG are numerous, its practical application in the medical domain also underscores ethical considerations given the critical nature of medical decision-making. Though RAG can help to constrain LLMs from generating incorrect output, a recent study has shown that even with RAG, LLMs may generate incorrect answers and explanations [48]. Any system that might suggest an incorrect treatment plan or diagnosis could have disastrous consequences for the patient. Therefore, the correctness of an LLM-based system is an ethical concern. As seen in Fig 3, 69 out of the 70 studies included in this review assessed the accuracy of their RAG pipeline, including the outputs generated by the model. The exception was [83], which focused on content relevance and suitability, evaluating whether the generated protocol section was specific to the disease and trial phase, among other criteria.

Data leakage is another known problem for LLMs that has been extensively studied [141–143]. Researchers have proposed RAG as a safer approach to reduce LLMs' tendency to output memorised data from its training. However, as argued in [144], information from pretraining/fine-tuning datasets (from the LLM) and the retrieval dataset can be potentially exposed when using RAG. In the context of healthcare, a retrieval dataset may contain sensitive patient information such as medications, diagnoses, and personally identifiable information (PII). As such RAG-based LLM system need to ensure and evaluate safety and privacy. The majority of the studies we reviewed do not address privacy issues with the exception of [79] and [103]. Li et al. [79] reduced privacy risks by removing identifiable information about the patient or doctor from their retrieval dataset. Another study used a multi-layered approach to ensure data privacy and security when using Azure OpenAI for sensitive healthcare data [103]. Another important concern that cannot be overlooked when LLMs are applied in the healthcare is safety. Only three out of the thirty-seven studies we reviewed evaluated their models against intentional or unintentional harms. In one study, adversarial prompting [145] was used to evaluate the robustness and reliability of system outputs in unexpected situations [24]. Another study used preoperative guidelines to guide decision-making of LLMs, thus, improve patients safety [73]. Similarly, a technique that flags safety concerns was developed, demonstrating zero instances of alarming red flags during testing [49].

Bias is another concern that is extensively studied in AI due to its ability to amplify inequity. Surprisingly, most of the papers we examined do not assess the existence of bias in the responses generated by their RAG systems. In the studies we reviewed that evaluated bias, Chen et al. [21] utilised a physician to determine the potential bias in the answers generated by DocQA, a RAG-enabled LLM. They observed reduced bias content with RAG compared

to an LLM alone. Though RAG substantially reduces the LLM's ability to use memorised data from model training [99], it does not avert bias completely, which is baked into the LLMs underlying training data. Various studies have indicated that RAG reduces hallucinations [19, 54,78,81,82,146]. However, different LLMs showed varying degrees of this effect. For instance, Aratesh et al. [54] found that GPT-4 exhibited the fewest hallucinations in their study.

Given LLMs high output variability, poor inherent explainability, and the risk of hallucinations, LLM-based healthcare applications that serve a medical purpose face challenges for approval as medical devices under US and EU laws [147], including the recently passed EU AI Act [148]. Consequently, ethical considerations such as bias, privacy, hallucination and safety are of paramount importance and should be addressed when working with RAG-based LLMs in healthcare. In a nutshell, these concerns can be addressed by implementing robust data privacy measures, promoting transparency and accountability, mitigating bias, emphasizing human oversight, and promoting human-autonomy. From our review, we see that only a few articles tackle these issues, and we see here wide margin of improvement.

## Data analysis

Fig 4a presents the distribution of languages of common retrieval datasets used for RAG in medical domain. It is evident that the majority (70.8%) of the datasets are in English, except for four datasets which are in Chinese. As seen in Fig 4b, the majority of the studies that employed RAG in the healthcare setting made use of proprietary models such as OpenAI's GPT-3.5/4 models. However, the use of these models raises privacy issues, especially when dealing with sensitive information such as patient data. In such cases, open-source models deployed in a controlled environment may be a suitable solution. However, it is worth noting that open-source models generally have lower performance compared to proprietary models and have a more limited context window [149]. A recent study comparing open-source and proprietary models, has shown that RAG with proprietary GPT-4o as the backbone LLM outperforms all others, demonstrating its superior adaptability with KG integration [114]. We also observed that the majority of the studies we reviewed made use of OpenAI's text-embedding-ada-002 model [150] for generating embeddings. Other embedding models used included BAAI general embedding (BGE) [67,89], and HuggingFace's all-MiniLM-L6-v2 [22]. Other studies used custom embedding models, as in [23]. In other studies, embedding fine-tuned was employed [91,117]. For example, Pang et al. [91] have shown that compared to direct vector retrieval, embedding fine-tuning allows for accurate capturing of similarity between a query and its relevant document, improving retrieval accuracy. A study by Kang et al. [72] demonstrated the feasibility of embeddings-free RAG in the medical domain. This suggests that one may not always need to utilise vector embeddings for successful RAG implementation.

The majority of the studies we surveyed used dense passage retrievers as a retrieval method in their RAG architecture. Few studies have used custom retrievers. For pre-retrieval, strategies employed are enhancing data granularity, adding metadata, optimising indexing, mixed retrieval and alignment optimisation. This involves methods such as chunking, knowledge graphs, creating Table-of-Contents and entity masking. Finally, studies have explored different modalities to improve the performance of their RAG architectures, ranging from one-shot learning, chain-of-thought and prompting to more advanced techniques such as using LLMs as agents. Prompting can impact the results of LLM outputs [55]. Therefore, exploring various prompting techniques and choosing the appropriate prompting strategy can mitigate errors and potential risks.

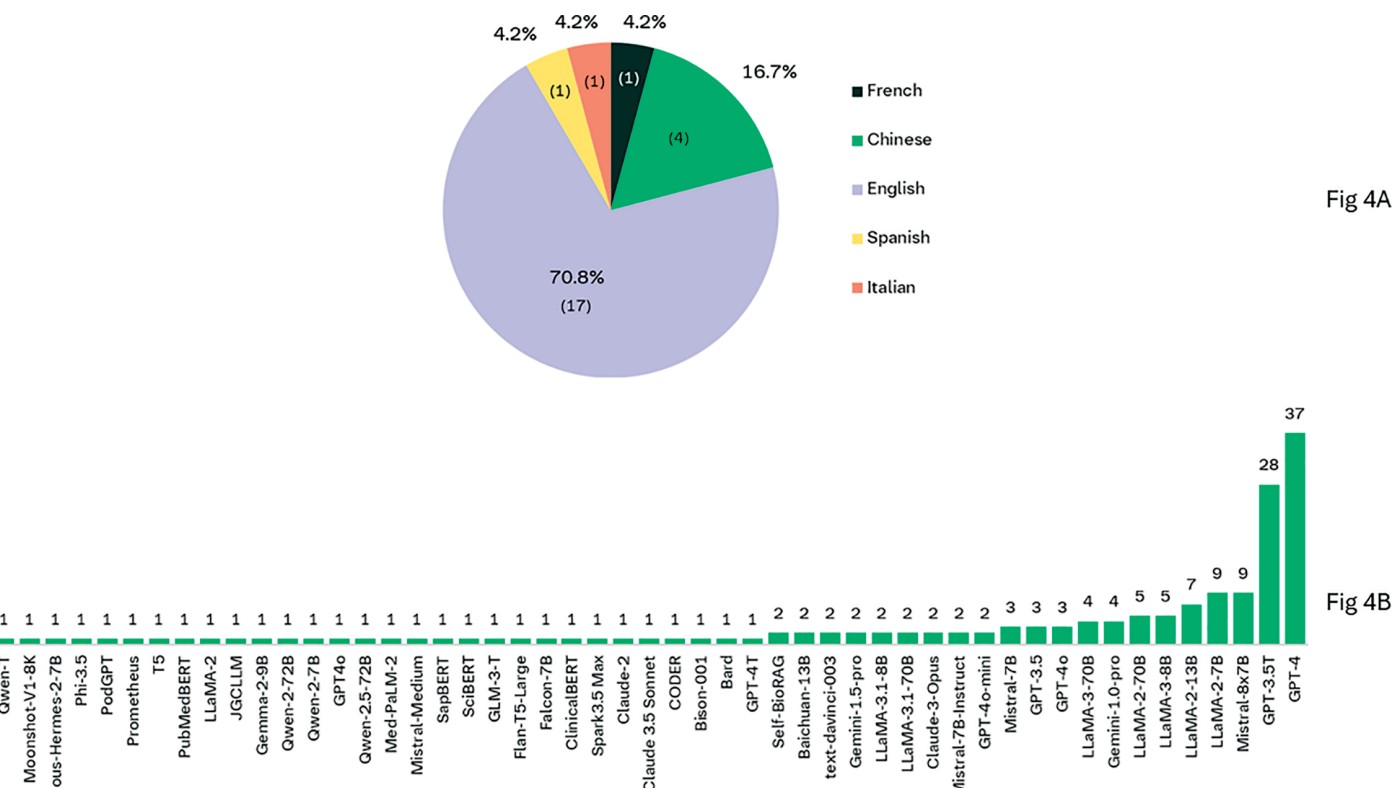

**Fig 4. a) Dataset Languages.** b) LLMs explored for retrieval-augmented generation in healthcare. Please note that some studies used more than one model, hence the total count of models is higher than the number of studies included in this review.

## Discussion

We comprehensively reviewed recent advancements of RAG-based approaches for LLMs in healthcare. Our survey discusses broader applications of RAG-based approaches within the landscape of LLMs for healthcare, dividing RAG approaches into three paradigms: naive, advanced and modular RAG. Moreover, we outline evaluation frameworks, including objectives and metrics used to assess the performance of RAG-based LLMs for healthcare.

Our findings indicate that proprietary LLMs such as GPT models are the most commonly used for RAG, employed in 53 out of the 70 studies we reviewed (see Table 3). The dominance of proprietary models is not surprising, given their superior performance in tasks like "zero-shot reasoning" when compared to open-source models. For example, GPT-4 and Claude 2 have consistently outperformed their open-source counterparts in these tasks [149], illustrating the strong capabilities and potential for off-the-shelf LLMs in solving complex problems in medicine. Another key observation from our review is the language bias in the datasets used as external knowledge sources. We found that the majority of these datasets are in English, with only a few exceptions in Chinese. This language bias presents a challenge in evaluating the performance of RAG on non-English datasets, other than Chinese. The lack of representative datasets highlight a gap in the current research and underscores the need for more diverse language datasets. Representation is crucial for fairness and equity in medical AI systems [151,152].

We found that most RAG-based studies are primarily focused on optimising retrieval and generation, including techniques such as incorporating metadata, re-ranking, chunking strategies, summarisation, and adapting prompts. However, we believe that these optimisation techniques, while important, may only lead to marginal improvements in performance. They may also prove inadequate when handling complex medical queries, which often require reasoning over the evidence. An example of this is answering multi-hop queries where the system must retrieve and reason over several pieces of supporting evidence [153]. For multi-hop document fetching, RAG on KGs enables fetching document nodes that are two or three hops away, which is suited for complex medical problems. Additionally, RAG studies in the medical domain typically employ a conventional architecture with a single round of retrieval. While iterative data refinement has been attempted for LLM training [154] and RAG in general domains [155,156], these methods have not yet been explored in the medical field. Recently, Xiong et al. [157] introduced the first approach and evaluations for incorporating follow-up queries in RAG specifically for medicine. Incorporating flexible information retrieval approaches may be necessary.

We observed that even within RAG, LLMs may not always respond as expected. This can be attributed to several factors: 1) the information contained in the pre-training of the model may leak in the final model answer; 2) irrelevant information may be retrieved, leading to inaccurate responses; 3) LLM generation can be unpredictable, resulting in unexpected outputs. Also, our findings indicate a lack of standardisation in the evaluation metrics used across different studies (as seen in Fig 3). This makes it difficult to compare the performance of the RAG systems across different studies, highlighting a need for more uniform evaluation metrics. Recently, a standard benchmark collection for evaluating clinical language understanding tasks was proposed [158]. Lastly, we find that the majority of the studies we reviewed do not address safety concerns. Ensuring safety is essential to ensure that the system causes no harm, protect patients' privacy and comply with regulations.

We acknowledge that our study is not exempt from limitations. First, we only included papers published in English, which may have excluded relevant studies in other languages. Second, keywords used in the search may have excluded some RAG-based studies in healthcare because RAG terminologies continues to rapidly evolve. Third, we relied on the information reported by the authors of the original papers, which may have introduced some bias or errors in our analysis. Last, despite conducting a systematic review we might have missed some relevant studies because we limited our search to studies published between January 2020 - March 2024.

Despite RAG advancements, retrieval quality remains a challenge. Several issues can arise from poor retrieval quality from external knowledge sources. For example, when not all the relevant chunks are retrieved (low context recall), it is challenging for the LLM to produce complete and coherent text. Also, the retrieved chunks might not align with the user query, potentially lead to hallucinations. To identify and mitigate hallucinations, different methods have been proposed. The authors in [5] proposed Med-Halt, a domain specific benchmark to evaluate and reduce hallucination in LLMs. Other researchers have proposed overcoming hallucination by using human-in-the-loop, algorithmic corrections and fine-tuning [159]. However, the aforementioned studies do not specifically focus on RAG scenarios. Wu et al. [160] proposed a high quality manually annotated dataset called RAGTruth and achieved comparative performance on existing prompt based techniques using SOTA LLMs, for e.g, GPT-4. Further research should focus on retrieval issues, by developing novel methods to find the most relevant information for the query. In addition, curation of diverse benchmark datasets for hallucination detection in healthcare, going beyond multi-choice questions and more inline with clinical practise, should constitute a primary research endeavour.

Current LLMs are limited by their context window. The context window determines the number of tokens the model can process and generate at a given user session. A right balance should be sought by the user, who should provide a sufficient context to the model without exceeding its context length [161]. Inadequate context can lead to a lack of necessary information, while excessive irrelevant context can impair the ability to recall relevant context. Ongoing research is exploring the benefits of longer context in enabling models to access more information from external knowledge sources [162]. This is especially important in healthcare settings, where clinicians often rely on longitudinal data, such as clinical notes, lab values, and imaging data. To incorporate multi-modal data effectively, LLMs need to consider much longer contexts. Future work should prioritise evaluating the impact of longer context on LLMs for healthcare. Though a longer context window can enhance the properties of RAG, it is not clear how this can reduce hallucination. A recent study has demonstrated that even with RAG, out of all the medical queries, approximately 45% of the responses provided by GPT-4 were not completely backed by the URLs retrieved [163]. Future studies should focus on techniques to robustly handle noise, information integration and improving the validity of sources, e.g., using post-hoc citation-enhanced generation [164].

The emergence of multimodal LLMs that can understand and output text, images, and audio presents exciting prospects for RAG. In healthcare, the gap in visual semantics and language understanding has been addressed by vision-language models that correlate visual semantics from medical images and text from medical reports or EHRs. For instance, Liu et al. [165] proposed contrastive language-image pre-training using zero-shot prompting to provide additional knowledge to help the model making explainable and accurate diagnosis from medical images. Beyond images, multimodal LLMs grounded in an individual's specific data to estimate disease risk have been proposed [20]. Other researchers have proposed RAG driven frameworks to improve multimodal EHRs representation [166].

Human evaluations remains crucial in assessing the output generated by RAG-based models. While human evaluation remains important, automated solutions are being developed to improve evaluation of LLMs by assessing both the output generated and information used to generate the answers. For instance, the RAGAs framework [135] allows to evaluate both generator and retriever separately. Automated benchmarks have also been proposed, integrating the evaluation of the ability of the system to retrieve and summarise relevant information [43]. Future research should explore combining automated metrics and human evaluation, ensuring that there is alignment. Finally, there is a need to shift from generic evaluation frameworks and benchmarks to contextual standardised evaluation metrics for RAG-based LLMs in healthcare.

While RAG holds immense promise in healthcare, its adoption must be guided by ethical principles, balancing innovation with patient privacy and safety. For instance, when external databases are accessed in RAG systems, there is a risk of inadvertently revealing sensitive information such as patient prescription information [79]. One way to overcome this challenge is to ensure that retrieval databases do not contain personally identifiable patient information. Additionally, composite structured prompting can be used to effectively extract retrieval data and evaluate privacy leakages by comparing LLM-generated outputs with the retrieved information [144]. Future studies, should explore novel measures to effectively overcome retrieval information leakages. Furthermore, it is crucial for LLM developers in healthcare to proactively address ethical issues throughout the AI development life cycle [167]. This proactive approach would foster trust and encourage the adoption of RAG-based LLMs in critical sectors such as healthcare [168,169]. Finally, if developed ethically, LLMs in medicine

can potentially increase access and equity in healthcare, for example enabling clinical trials to be more inclusive or providing treatments that are tailored to patients from diverse demographics.

## Conclusion

In this paper, we comprehensively outline RAG's advancement in grounding and improving the capabilities of LLMs in the medical domain. First, we discuss the available datasets used for grounding LLMs for healthcare tasks such as question-answer/dialogue and information retrieval. Second, we compare the models, and the retrieval and augmentation techniques employed by existing studies. Third, we assess evaluation frameworks proposed for RAG systems in the medical domain. Our results shows that there is a growing interest in applying RAG to ground LLMs in healthcare, and proprietary LLMs are the most commonly used models. When it comes to evaluation of RAG-pipeline, our findings highlight the absence of a standardised framework for assessing RAG pipelines in the medical field. Despite these challenges, RAG has the potential to ground and customise the domain knowledge of LLMs for healthcare, by integrating dynamic data, standards, and a complete integration with individual scenarios. Therefore, revolutionise various areas in healthcare from drug development to disease prediction and personalised care management. Nevertheless for RAG to be effectively implemented in healthcare, it is essential to adequately address challenges such as integration of information, handling of noise, source factuality and ethical considerations. Finally, with continuous improvements LLMs will play an instrumental role in shaping the future of healthcare, driving innovation and enhancing patient care.

## Supporting information

**S1 File. This file contains the raw data of all papers collected.**
(XLSX)

**S2 File. This file includes the data extracted from the papers.**
(XLSX)

**S3 File. This file provides the evaluation metrics used to assess the performance of RAG pipelines in the various papers.**
(XLSX)

## Acknowledgments

We would like to express our deepest gratitude to reviewers for reviewing our work. Their insightful comments and feedback has greatly improved our manuscript.

## Author contributions

**Conceptualization:** Pietro Mascheroni, Jan Seidel.

**Data curation:** Lameck Mbangula Amugongo.

**Formal analysis:** Lameck Mbangula Amugongo.

**Investigation:** Lameck Mbangula Amugongo, Pietro Mascheroni.

**Methodology:** Lameck Mbangula Amugongo, Pietro Mascheroni, Steven Brooks.

**Visualization:** Lameck Mbangula Amugongo.

**Writing – original draft:** Lameck Mbangula Amugongo, Pietro Mascheroni, Steven Brooks, Stefan Doering, Jan Seidel.

**Writing – review & editing:** Lameck Mbangula Amugongo, Pietro Mascheroni, Steven Brooks, Stefan Doering, Jan Seidel.

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
