## [Decision Letter · Decision Letter 0]

PDIG-D-24-00337Retrieval Augmented Generation for Large Language Models in healthcare: A systematic reviewPLOS Digital Health Dear Dr. Amugongo, Thank you for submitting your manuscript to PLOS Digital Health. After careful consideration, we feel that it has merit but does not fully meet PLOS Digital Health's publication criteria as it currently stands. Therefore, we invite you to submit a revised version of the manuscript that addresses the points raised during the review process. Please submit your revised manuscript within 60 days Apr 05 2025 11:59PM. If you will need more time than this to complete your revisions, please reply to this message or contact the journal office at digitalhealth@plos.org. Please include the following items when submitting your revised manuscript:* A rebuttal letter that responds to each point raised by the editor and reviewer(s). You should upload this letter as a separate file labeled 'Response to Reviewers'. This file does not need to include responses to any formatting updates and technical items listed in the 'Journal Requirements' section below.* A marked-up copy of your manuscript that highlights changes made to the original version. You should upload this as a separate file labeled 'Revised Manuscript with Track Changes'.* An unmarked version of your revised paper without tracked changes. You should upload this as a separate file labeled 'Manuscript'. If you would like to make changes to your financial disclosure, competing interests statement, or data availability statement, please make these updates within the submission form at the time of resubmission. Guidelines for resubmitting your figure files are available below the reviewer comments at the end of this letter. We look forward to receiving your revised manuscript. Kind regards, Laura Sbaffi, PhD, MA, MScSection EditorPLOS Digital Health Laura SbaffiSection EditorPLOS Digital Health Leo Anthony CeliEditor-in-ChiefPLOS Digital Healthorcid.org/0000-0001-6712-6626 **Journal Requirements:**

1. As required by our policy on Data Availability, please ensure your manuscript or supplementary information includes the following: 

**Additional Editor Comments (if provided):****Reviewers' Comments:** Reviewer's Responses to Questions

**Comments to the Author**

1. Does this manuscript meet PLOS Digital Health’s publication criteria? Is the manuscript technically sound, and do the data support the conclusions? The manuscript must describe methodologically and ethically rigorous research with conclusions that are appropriately drawn based on the data presented.

Reviewer #1: Yes

Reviewer #2: Yes

2. Has the statistical analysis been performed appropriately and rigorously?

Reviewer #1: N/A

Reviewer #2: Yes

3. Have the authors made all data underlying the findings in their manuscript fully available (please refer to the Data Availability Statement at the start of the manuscript PDF file)?

Reviewer #1: Yes

Reviewer #2: Yes

4. Is the manuscript presented in an intelligible fashion and written in standard English?

Reviewer #1: Yes

Reviewer #2: Yes

5. Review Comments to the Author

Reviewer #1: In this study, the authors performed a systematic review of RAG applications with respect to LLMs in healthcare. This is a timely and very good study, well-structured, and clearly presented. My comments are only in the direction of improving the manuscript.

COMMENTS:

Comment 1: The notation of “Naïve RAG” which is used by the authors is not very common in the literature. Could the author explain why they call it Naïve RAG? I think this should be changed.

Similar story of the term “Advanced RAG”. How do you define “Advanced”?

Comment 2: Given the very fast-paced nature of this domain, especially RAG in the past months, I propose the authors to consider the literature up until now (September 2024) instead of March 2024, as there has been further work published recently such as (1-3).

Comment 3: Table 1 could be integrated into the text and removed from the table list.

Comment 4: Have the authors obtained permission from the owner of Figure 2 to include it in the manuscript?

Furthermore, the fonts are very small in that figure and not easily readable.

Comment 5: A better representation could be chosen instead of Table 5. Currently, it is difficult to read through it.

Comment 6: Figure 3 could be improved:

- Fonts to be larger

- subfigure A could be integrated into subfigure b.

References:

(1) Freyer, Oscar, et al. "A future role for health applications of large language models depends on regulators enforcing safety standards." The Lancet Digital Health 6.9 (2024): e662-e672.

(2) Arasteh, Soroosh Tayebi, et al. "RadioRAG: Factual Large Language Models for Enhanced Diagnostics in Radiology Using Dynamic Retrieval Augmented Generation." arXiv preprint arXiv:2407.15621 (2024).

(3) Luo, Ming-Jie, et al. "Development and evaluation of a retrieval-augmented large language model framework for ophthalmology." JAMA ophthalmology (2024).

Reviewer #2: This is a systematic review on RAG-LLM in healthcare, with a focus on technical architecture, dataset characteristics and ethical considerations. This study provides a much needed review on the different RAG methodologies, systematic bias such as the dominance of English and Chinese datasets in studies, the underdevelopment of standardized evaluation frameworks, and the general omission of ethical considerations in RAG applications in healthcare. There has not been a systematic review published of this nature and I rate the novelty as high.

The systematic review follows the PRIMSA guidelines, with appropriate search criteria, inclusion and exclusion criteria.

Comments:

1) It is not clear who performed the screening of the articles, if independent screening was performed.

2) A quantitative analysis of the performance of RAG vs native LLMs will be helpful in contextualising the utility of RAG

6. PLOS authors have the option to publish the peer review history of their article (what does this mean?). If published, this will include your full peer review and any attached files.

**Do you want your identity to be public for this peer review?** For information about this choice, including consent withdrawal, please see our Privacy Policy.

Reviewer #1: No

Reviewer #2: No

---

## [Decision Letter · Decision Letter 1]

Retrieval Augmented Generation for Large Language Models in healthcare: A systematic review

PDIG-D-24-00337R1

Dear Dr. Mbangula Lameck Amugongo,

We are pleased to inform you that your manuscript 'Retrieval Augmented Generation for Large Language Models in healthcare: A systematic review' has been provisionally accepted for publication in PLOS Digital Health.

Best regards,

Xiaoli Liu, PhD

Academic Editor

PLOS Digital Health

**Additional Editor Comments (if provided):**

Dear Dr. Mbangula Lameck Amugongo,

Thank you for your patience! We have received the feedback and are pleased to inform you that your revisions have addressed the reviewers' comments effectively. I hope you will further check the contents, verify the results and conclusions, and ensure there are no errors. We look forward to publishing your work as soon as possible.

Best regards,

Dr. Liu

**Reviewer Comments (if any, and for reference):**

Reviewer's Responses to Questions

**Comments to the Author**

1. If the authors have adequately addressed your comments raised in a previous round of review and you feel that this manuscript is now acceptable for publication, you may indicate that here to bypass the “Comments to the Author” section, enter your conflict of interest statement in the “Confidential to Editor” section, and submit your "Accept" recommendation.

Reviewer #1: All comments have been addressed

2. Does this manuscript meet PLOS Digital Health’s publication criteria? Is the manuscript technically sound, and do the data support the conclusions? The manuscript must describe methodologically and ethically rigorous research with conclusions that are appropriately drawn based on the data presented.

Reviewer #1: Yes

3. Has the statistical analysis been performed appropriately and rigorously?

Reviewer #1: N/A

4. Have the authors made all data underlying the findings in their manuscript fully available (please refer to the Data Availability Statement at the start of the manuscript PDF file)?

Reviewer #1: Yes

5. Is the manuscript presented in an intelligible fashion and written in standard English?

Reviewer #1: Yes

6. Review Comments to the Author

Reviewer #1: I thank the authors for addressing my original comments.

7. PLOS authors have the option to publish the peer review history of their article (what does this mean?). If published, this will include your full peer review and any attached files.

**Do you want your identity to be public for this peer review?** For information about this choice, including consent withdrawal, please see our Privacy Policy.

Reviewer #1: No
